# Crank Nicholson scheme to examine the fractional-order unsteady nanofluid flow of free convection of viscous fluids

Tamour Zubair[1]*, Muhammad Usman[2], Kottakkaran Sooppy Nisar[3], Ilyas Khan[4], Madiha Ghamkhar[5], Muhammad Ahmad[5]

**1** School of Mathematical Sciences, Peking University, Beijing, China, **2** Department of Mathematics, National University of Modern Languages (NUML), Islamabad, Pakistan, **3** Department of Mathematics, College of Arts and Sciences, Wadi Aldawaser, Prince Sattam Bin Abdulaziz University, Al-Kharj, Saudi Arabia, **4** Department of Mathematics, College of Science Al-Zulfi, Majmaah University, Al-Majmaah, Saudi Arabia, **5** University of Agriculture, Faisalabad, Pakistan

* tamourzubair@pku.edu.cn

**Data Availability Statement:** All relevant data are within the paper.

**Funding:** The author(s) received no specific funding for this work.

## Abstract

Fractional fluid models are usually difficult to solve analytically due to complicated mathematical calculations. This difficulty in considering fractional model further increases when one considers nth order chemical reaction. Therefore, in this work an incompressible nanofluid flow as well as the benefits of free convection across an isothermal vertical sheet is examined numerically. An nth order chemical reaction is considered in the chemical species model. The specified velocity (wall's) is time-based, and its motion is translational into mathematical form. The fractional differential equations are used to express the governing flow equations (FDEs). The non-dimensional controlling system is given appropriate transformations. A Crank Nicholson method is used to find solutions for temperature, solute concentration, and velocity. Variation in concentration, velocity, and temperature profiles is produced as a result of changes in discussed parameters for both Ag-based and Cu-based nanofluid values. Water is taken as base fluid. The fractional-order time evaluation has opened the new gateways to study the problem into a new direction and it also increased the choices due to the extended version. It records the hidden figures of the problem between the defined domain of the time evaluation. The suggested technique has good accuracy, dependability, effectiveness and it also cover the better physics of the problem specially with concepts of fractional calculus.

## 1. Introduction

Natural convection process is that type of flow situation in which a liquid such as water as an example of Newtonian fluid, in which the fluid motion is produced by an external source instead by some parts of the fluid being heavier than other parts. More exactly, it is a specific kind of self-persistent flow with a high-temperature gradient, as a natural convection flow. This factor is then endorsed in order to get the non-uniform density. Because of changes in

**Competing interests:** The authors have declared that no competing interests exist.

density and gravitational field, buoyancy effects promote current movement. The aforementioned occurrences occur often in nature and have been documented in a variety of technical and engineering settings [1]. The most common model in convective flow models is natural convection, which involves the movement of heat and mass near a moving sheet. The aforementioned concept is often used in solar energy collectors, nuclear reactor architecture, and electronic devices. Various writers discussed the wonders of natural convection with the transfer of heat and mass, as well as the Newtonian/Non-Newtonian character of the fluid, due to its wide range of possible uses. The effect of convection (or free convection) on the accelerating plate in a perpendicular position was examined in [2], where they utilized the Laplace transformation technique (LTM) to examine the solution for two distinct circumstances, namely, the constant heat flux and isothermal plate. Refer to [3] for a free-convection flow issue in which a vertical plate is constructed in such a manner that it increases exponentially. Some of the prospective information may be examined methodically in references [4–11].

Following the contribution of Choi [12], the discipline of fluid mechanics received valuable concertation. Because he focused on thermal conductivity enrichment ideas related to fluids, he created the nano-fluids discipline. He demonstrated that nanoparticles, which are microscopic and small particles, may be put into fluids to convert them into nanofluids. The extensive research and experimental results revealed that it improves the thermal characteristics of the conventional fluid. As a result of this dramatic modernization, this area acquired considerable significance, and a large body of work is accessible in the literature. Sheikholelsami and Ganji [13] investigated the convective heat transport of nanofluids. [14–32] provide a comprehensive examination of nano-fluids and applications from different perspectives.

Previously, the fractional calculus theory sparked widespread interest due to its wide range of applications in physics and engineering [10]. This kind of research has made use of multidimensional dynamics such as wave, viscoelastic, and relaxation activities. Because of the operators, we developed a straightforward method for introducing fractional ordered derivatives into linear viscous models, which drew much attention to this area.

This research looked at the physical elements of the issue of fractional-order derivatives between certain domains. The fractional calculus makes visible contributions to various technical and scientific circumstances, including neurology, capacitor theory, viscoelasticity, electro-analytical chemistry, and electrical circuits [33,34]. Several authors [35–39] suggested several techniques for dealing with the nonlinearity of fractional differential equations. Despite the fact that there is extensive research literature on fluid flows, numerous mathematical and fluid models are developed and effectively solved using the fractional calculus method; see, for example, some useful investigations in this direction [7,40–42].

The above literature shows that several investigations are done on convection heat transfer using classical/fractional models [43–47] and [48–53]. However, in all these models, particularly those they are involved with fractional derivatives, no attention is given to nth order chemical reaction in species concentration and the free convection flow of viscous nanofluid using fractional derivatives. Therefore, the main objective of this work is to fill this gap. More exactly, in this article, water-based nanofluid is considered with Ag and Cu nanoparticles. The fractional differential equations are used to express the governing flow, heat, and species concentration equations. The Crank Nicholson technique is used to generate numerical results [54]. Variation in numerical implications of concentration, velocity, and temperature profiles is shown as a result of variations in various parameters for both Ag-based and Cu-based nanofluid values. The collected findings demonstrate the suggested technique's accuracy, dependability, and effectiveness.

## 2. Mathematical and geometrical analysis

Consider the mass and energy (heat) transmission performance of a nanofluid that is unsteady free convection, incompressible, one dimensional, viscous, and radiative, and is limited among specified plates that are parallel, filled with a porous material, and have distance $d$. Initially ($t = 0$), the fluid and plates are assumed to be stationary, and $T_\infty$ and $C_\infty$ are the constant temperature and constant concentration, respectively. For $t > 0$, the heat transfer process and surface temperature are proportionate. The nth-order chemical reaction is taken into account. Flow may be described using the following partial differential equations in light of the Boussinesq approximation.

$$\frac{\partial u}{\partial t} = v_{nf}\frac{\partial^2 u}{\partial y^2} + \frac{g}{\rho_{nf}}\beta_{nf}(T - T_\infty) + \frac{1}{\rho_{nf}}(\boldsymbol{J} \times \boldsymbol{B})_x - \frac{v_{nf}\phi_m}{k}u, \tag{1}$$

$$(\rho C_P)_{nf}\frac{\partial T}{\partial t} = k_{nf}\frac{\partial^2 T}{\partial y^2} - \frac{\partial q_r}{\partial y} \tag{2}$$

$$\frac{\partial C}{\partial t} = D\frac{\partial^2 C}{\partial y^2} + \gamma(C - C_\infty)^n, \tag{3}$$

Where $u(y, t)$, $T(y, t)$, $C(y, t)$, $g$, $v_{nf}$, $\rho_{nf}$, $\beta_{nf}$, $\sigma_{nf}$, $(C_P)_{nf}$, $k_{nf}$ are the velocity, temperature, concentration gravitational acceleration, kinematics viscosity, density, heat transfer constant, electrical conductivity, heat capacity, and thermal conductivity. $\boldsymbol{J}$, $D$, $\gamma$, $n$ and $\phi_m$ are the parameters for current density, mass diffusion, rate of chemical reaction, order of chemical reaction, and porosity, respectively.

$$v_{nf} = \frac{\mu_{nf}}{\rho_{nf}}, \mu_{nf} = \frac{\mu_f}{(1-\phi)^{2.5}}, \rho_{nf} = \rho_f\left((1-\phi) + \phi\frac{\rho_s}{\rho_f}\right), (\rho\beta)_{nf} = (1-\phi)(\rho\beta)_f +$$

$$\phi(\rho\beta)_s, \sigma_{nf} = \sigma_f\left(1 + \frac{3(\sigma-1)\phi}{\sigma+2-(\sigma-1)\phi}\right), \sigma = \frac{\sigma_s}{\sigma_f}, k_{nf} = k_f\frac{k_s + 2k_f - 2\phi\left(k_f - k_s\right)}{k_s + 2k_f + \phi\left(k_f - k_s\right)} \tag{4}$$

Where in Eqs (1)–(4), $\rho_f$, $\rho_s$, $\beta_f$, $\beta_s$, $\mu_{nf}$, $\mu_f$, $\sigma_{nf}$, $\sigma_f$, $\sigma_s$, $k_f$, $k_s$ and $\phi$ are density, the density of solid particle, heat transfer constant, heat transfer constant for solid particle, viscosity, viscosity, viscosity of solid particle, electrical conductivity, the electrical conductivity of solid, thermal conductivity, thermal conductivity for solid, where the subscripts $nf$ and $f$ are for nanofluid and fluid, respectively. The current density value is

$$\boldsymbol{J} = \sigma_{nf}(E + \boldsymbol{V} \times \boldsymbol{B}), \tag{5}$$

where $E$ is the electric field. Cogley et al. shows that [30]:

$$\frac{\partial q_r}{\partial y} = 4(T - T_\infty)\int_0^\infty k_{\lambda_w}\left(\frac{de_{b\lambda}}{dt}\right)_w d\lambda, \tag{6}$$

Where $k_\lambda$, $e_{b\lambda}$, $w$ are the absorption coefficient, plank function and value at the wall $y = d$.

Substituting the values from Eqs (4–6) into Eqs (1–3), once obtained

$$\frac{\partial u}{\partial t} = v_{nf}\frac{\partial^2 u}{\partial y^2} + \frac{g}{\rho_{nf}}\beta_{nf}(T - T_\infty) - \frac{\sigma_{nf}B_0^2}{\rho_{nf}}u - \frac{v_{nf}\phi_m}{k}u, \tag{7}$$

$$(\rho c_P)_{nf}\frac{\partial T}{\partial t} = k_{nf}\frac{\partial^2 T}{\partial y^2} - 4(T - T_\infty)I, \tag{8}$$

$$\frac{\partial C}{\partial t} = D\frac{\partial^2 C}{\partial y^2} + \gamma(C - C_\infty)^n, \tag{9}$$

where $I = \int_0^\infty k_{\lambda_w}\left(\frac{de_{b\lambda}}{dt}\right)_w d\lambda$.

The associated initial and boundary conditions for Eqs (7)–(9) are:

$$u(y, 0) = 0, T(y, 0) = T_\infty, C(y, 0) = C_\infty.$$

for $t > 0$, $u(0, t) = u(1, t) = 0, \frac{\partial T}{\partial y}\big|_{y=0} = -\frac{1}{d}T(0, t), T(1, t) = T_\infty, C(0, t) = C_\infty, C(1, t) = C_w$.

To nondimensionalize the above system of PDEs let us consider the following transformations:

$$u = \frac{v_f}{d}\bar{u}, y = d\bar{y}, t = \frac{d^2}{v_f}\bar{t}, C = C_\infty\bar{C} + C_\infty, T = T_\infty\bar{T} + T_\infty,$$

Using the above transformation into (7–9), once obtained

$$\frac{\partial\bar{u}}{\partial\bar{t}} = \frac{1}{A(1-\phi)^{2.5}}\frac{\partial^2\bar{u}}{\partial\bar{y}^2} + \frac{BGr}{A}\bar{T} - \frac{M^2 C_1}{A}\bar{u} - \frac{N}{(1-\phi)^{2.5}AD_1}\bar{u}, \tag{10}$$

$$EPr\frac{\partial\bar{T}}{\partial\bar{t}} = D_1\frac{\partial^2\bar{T}}{\partial\bar{y}^2} - R\bar{T}, \tag{11}$$

$$\frac{\partial\bar{C}}{\partial\bar{t}} = \frac{1}{Le}\frac{\partial^2\bar{C}}{\partial\bar{y}^2} + \delta\bar{C}^n, \tag{12}$$

Moreover $A, B, C_1, D_1, E, F$ are just constants which introduced for simplicity and is given by:

$$A = \left((1-\phi) + \phi\frac{\rho_s}{\rho_f}\right), B = \left((1-\phi) + \frac{\phi(\rho\beta)_s}{(\rho\beta)_f}\right), C_1 = \left(1 + \frac{3(\sigma-1)\phi}{\sigma+2-(\sigma-1)\phi}\right),$$

$$D_1 = \frac{k_s + 2k_f - 2\phi\left(k_f - k_s\right)}{k_s + 2k_f + \phi\left(k_f - k_s\right)}, E = \left[(1-\phi) + \phi\frac{(\rho c_P)_s}{(\rho c_P)_f}\right].$$

Transformed form of IC and BC's are given as:

$$\bar{u}(y,0) = \bar{T}(y,0) = \bar{C}(y,0) = 0,$$

$$\bar{u}(0,t) = 0, \frac{\partial \bar{T}}{\partial y}\bigg|_{y=0} = -\bar{T}(0,t) - 1, \bar{C}(0,t) = 0, \bar{u}(1,t) = 0, \bar{T}(1,t) = 0, \bar{C}(1,t) = 1.$$

The Caputo time fractional form of Eqs (8) and (9) are explained as follow also replacing $\bar{t}, \bar{y}, \bar{u}, \bar{T}$ and $\bar{C}$ by $t, y, u, T$ and $C$:

$$D_t^\alpha u(y,t) = \frac{1}{A(1-\phi)^{2.5}}\frac{\partial^2 u}{\partial y^2} + \frac{BGr}{A}T - \frac{M^2 C_1}{A}u - \frac{N}{AD_1}u, \tag{13}$$

$$EPrD_t^\alpha T(y,t) = D_1 \frac{\partial^2 T}{\partial y^2} - RT, \tag{14}$$

$$D_t^\alpha C(y,t) = \frac{1}{Le}\frac{\partial^2 C}{\partial \bar{y}^2} + \delta C^n. \tag{15}$$

where $D_t^\alpha V(y,t)$
$$\begin{cases} \dfrac{1}{\Gamma(1-\alpha)}\displaystyle\int_0^t \dfrac{1}{(t-\tau)^\alpha}\dfrac{\partial V(y,\tau)}{\partial \tau}d\tau, 0 < \alpha < 1, \\ \dfrac{\partial V(y,t)}{\partial t}, \alpha = 1. \end{cases}$$

## 3. Finite difference scheme

Crank Nicolson method (CNM) is projected to construct the numerical solution of problem (13–15) in this section. Consider the problem (13–15) for $n = 1$:

$$\frac{\partial^\alpha}{\partial t^\alpha}u(y,t) = \frac{1}{A(1-\phi)^{2.5}}\frac{\partial^2 u}{\partial y^2} + \frac{BGr}{A}T(y,t) - \frac{M^2 C_1}{A}u(y,t) - \frac{N}{AD_1}u(y,t), \tag{16}$$

$$E\,Pr\frac{\partial^\alpha}{\partial t^\alpha}T(y,t) = D_1\frac{\partial^2 T}{\partial y^2} - RT(y,t), \tag{17}$$

$$D_t^\alpha C(y,t) = \frac{1}{Le}\frac{\partial^2 C}{\partial \bar{y}^2} + \delta C(y,t). \tag{18}$$

The boundary condition associated with above system given in above. In above $0 \le \alpha \le 1$ is Caputo derivative of fractional order. Consider that the above fractional-order system (16)–(18) has sufficiently smooth and has unique. Assume that $x_j = jh$, $0 \le j \le M$ with $Mh = 1$ and $t_n = n_\tau$, $0 \le n \le N$. Here $h$ and $\tau$ indicate the space and time step length, $M$ and $N$ are represents the number of grids point. Fractional order derivate can discretize as [34]:

$$D_t^\alpha Q(y,t) = \frac{1}{\tau^\alpha \Gamma(2-\alpha)}\left[Q_j^{n+1} - Q_j^n + \sum_{i=1}^n \left(Q_j^{n-i+1} - Q_j^{n-i}\right)\left((i+1)^{1-\alpha} - i^{1-\alpha}\right)\right] + O(\tau),$$

and the second-order derivative using Crank-Nicholson idea can be discretized as under:

$$\frac{\partial^2}{\partial y^2} Q(y,t) = \frac{1}{2h^2}\left[\left(Q_{j+1}^{n+1} - 2Q_j^{n+1} + Q_{j-1}^{n+1}\right) + \left(Q_{j+1}^n - 2Q_j^n + Q_{j-1}^n\right)\right] + O(h^2).$$

Using the above-discretized formulas, system (16)–(18) takes the following form:

$$-\omega_{\tilde{u}}\tilde{u}_{j+1}^{n+1} + (\vartheta_{\tilde{u}} + 2\omega_{\tilde{u}})\tilde{u}_j^{n+1} - \omega_{\tilde{u}}\tilde{u}_{j-1}^{n+1}$$
$$= \omega_{\tilde{u}}\tilde{u}_{j+1}^n + \left(\vartheta_{\tilde{u}} - 2\omega_{\tilde{u}} - \left(\frac{M^2 C_1}{A} + \frac{N}{(1-\phi)^{2.5} A D_1}\right)\right)\tilde{u}_j^n + \omega_{\tilde{u}}\tilde{u}_{j-1}^n + \frac{BGr}{A}\tilde{T}_j^n$$
$$- \vartheta_{\tilde{u}}\sum_{i=1}^n \left(\tilde{u}_j^{n-i+1} - \tilde{u}_j^{n-i}\right)b_i,$$

$$-\omega_{\tilde{T}}\tilde{T}_{j+1}^{n+1} + (\vartheta_{\tilde{T}} + 2\omega_{\tilde{T}})\tilde{T}_j^{n+1} - \omega_{\tilde{T}}\tilde{T}_{j-1}^{n+1}$$
$$= \omega_{\tilde{T}}\tilde{T}_{j+1}^n + (\vartheta_{\tilde{T}} - 2\omega_{\tilde{T}} - R)\tilde{T}_j^n + \omega_{\tilde{T}}\tilde{T}_{j-1}^n - \vartheta_{\tilde{T}}\sum_{i=1}^n \left(\tilde{T}_j^{n-i+1} - \tilde{T}_j^{n-i}\right)b_i,$$

$$-\omega_{\tilde{C}}\tilde{C}_{j+1}^{n+1} + (\vartheta_{\tilde{C}} + 2\omega_{\tilde{C}})\tilde{C}_j^{n+1} - \omega_{\tilde{C}}\tilde{C}_{j-1}^{n+1}$$
$$= \omega_{\tilde{C}}\tilde{C}_{j+1}^n + (\vartheta_{\tilde{C}} - 2\omega_{\tilde{C}} + \delta)\tilde{C}_j^n + \omega_{\tilde{C}}\tilde{C}_{j-1}^n - \vartheta_{\tilde{C}}\sum_{i=1}^n \left(\tilde{C}_j^{n-i+1} - \tilde{C}_j^{n-i}\right)b_i,$$

where $\omega_{\tilde{u}} = \frac{1}{A(1-\phi)^{2.5}}\frac{1}{2h^2}$, $\vartheta_{\tilde{u}} = \frac{1}{\tau^\alpha \Gamma(2-\alpha)}$, $\omega_{\tilde{T}} = \frac{1}{2h^2}D_1$, $\vartheta_{\tilde{T}} = \frac{EPr}{\tau^\alpha \Gamma(2-\alpha)}$, $\omega_{\tilde{C}} = \frac{1}{Le}\frac{1}{2h^2}$,

$$\vartheta_{\tilde{C}} = \frac{1}{\tau^\alpha \Gamma(2-\alpha)}, b_i = \left((i+1)^{1-\alpha} - i^{1-\alpha}\right).$$

$$\mathbf{A}_1\mathbf{v}^1 = \mathbf{B}\mathbf{v}^0 + \frac{BGr}{A}\mathbf{C}\mathbf{v}^0,$$

for $n \geq 1$,

$$\mathbf{A}_{n+1}\mathbf{v}^{n+1} = \mathbf{B}_{n+1}\mathbf{v}^n + \mathbf{s}_1^{n+1}\mathbf{v}^n + \mathbf{s}_2^{n+1}\mathbf{v}^{n-1} + \cdots + \mathbf{s}_n^{n+1}\mathbf{v}^1 + \mathbf{b}^{n+1}\mathbf{v}^0 + \frac{BGr}{A}\mathbf{C}\mathbf{v}^n.$$

In above $\mathbf{A}_{n+1}, \mathbf{B}_{n+1}, \mathbf{v}_n, \mathbf{s}_n^{n+1}, \mathbf{C}$ and $\mathbf{b}^{n+1}$ are represents the block matrices which are defined as follow:

$$\mathbf{A}_{n+1} = \begin{bmatrix} \mathbf{A}_{n+1}^{\tilde{u}} & \mathbf{O} & \mathbf{O} \\ \mathbf{O} & \mathbf{A}_{n+1}^{\tilde{T}} & \mathbf{O} \\ \mathbf{O} & \mathbf{O} & \mathbf{A}_{n+1}^{\tilde{C}} \end{bmatrix}, \mathbf{B}_{n+1} = \begin{bmatrix} \mathbf{B}_{n+1}^{\tilde{u}} & \mathbf{O} & \mathbf{O} \\ \mathbf{O} & \mathbf{B}_{n+1}^{\tilde{T}} & \mathbf{O} \\ \mathbf{O} & \mathbf{O} & \mathbf{B}_{n+1}^{\tilde{C}} \end{bmatrix}, \mathbf{C} = \begin{bmatrix} \mathbf{O} & \mathbf{I} & \mathbf{O} \\ \mathbf{O} & \mathbf{O} & \mathbf{O} \\ \mathbf{O} & \mathbf{O} & \mathbf{O} \end{bmatrix}, \mathbf{s}_n^{n+1}$$

$$= \begin{bmatrix} \mathbf{c}_n^T & \mathbf{O} & \mathbf{O} \\ \mathbf{O} & \mathbf{d}_n^T & \mathbf{O} \\ \mathbf{O} & \mathbf{O} & \mathbf{e}_n^T \end{bmatrix}^{n+1}, \mathbf{v}^n = \begin{bmatrix} \mathbf{u} \\ \mathbf{T} \\ \mathbf{C} \end{bmatrix}^n, \mathbf{b}^{n+1} = \begin{bmatrix} \mathbf{b}_n^T & \mathbf{O} & \mathbf{O} \\ \mathbf{O} & \mathbf{b}_n^T & \mathbf{O} \\ \mathbf{O} & \mathbf{O} & \mathbf{b}_n^T \end{bmatrix}^{n+1},$$

where the matrices $\mathbf{A}_{n+1}^{\tilde{u}}, \mathbf{A}_{n+1}^{\tilde{T}}, \mathbf{A}_{n+1}^{\tilde{C}}, \mathbf{B}_{n+1}^{\tilde{u}}, \mathbf{B}_{n+1}^{\tilde{T}}, \mathbf{B}_{n+1}^{\tilde{C}}, \mathbf{c}_n^T, \mathbf{d}_n^T, \mathbf{e}_n^T, \mathbf{I}$ and $\mathbf{b}_n^T$ present in [33,34] and

**u** and **T** are given as:

$$\mathbf{u}^n = \left[ u_1^n, u_2^n, u_3^n, \ldots, u_{M-2}^n, u_{M-1}^n \right]^T,$$

$$\mathbf{T}^n = \left[ T_1^n, T_2^n, T_3^n, \ldots, T_{M-2}^n, T_{M-1}^n \right]^T.$$

## 4. Discussion about numerical outcomes

The parametric study is provided to investigate the physics of the problem described in the preceding section. The fractional finite difference technique is used to find a numerical solution. Figs 1–11 show dimensionless velocity, temperature, and concentration plotted against the change of various parameters listed in Table 1.

Figs 1–5 depict the behaviour of velocity for water based nanofluid (with Prandtl number Pr = 6.2) containing copper ($Cu$) and silver ($Ag$) nanoparticles) for various values of fractional parameter $\alpha$, as well as Hartmann number ($M$), porosity parameter ($N$), Grashof number ($Gr$), time ($t$) and solid volume fraction ($\phi$) [43]. At time $t$ = 0.3, the decreasing behaviour of velocity for Hartmann number ($M^2$) and fractional parameter $\alpha$ is shown in Fig 1.

$M$ arises in the problem as a significance of substantial magnetic field effects, as is well known. As a consequence, magnetic forces working against the flow process become stronger, resulting in a decrease in velocity. Normally, this parameter causes the temperature to rise and the collision process to accelerate, which has a noticeable influence on the velocity, as seen in Fig 1. The fractional parameter stores the time evaluation values of velocity fluids, indicating that the velocity is steadily decreasing and nearing the fractional parameter's integer value. As a result, the fractional parameter traces the location of the fluid particles.

Fig 2 illustrates the effect of changing numerical values N and on velocity at $t$ = 0.3. As the value of the porosity parameter is increased, the velocity impact diminishes ($N$). Increased porosity implies an increase in the degree of resistance. That is why the velocity of nanofluid decreases as the porosity parameter increases ($N$).

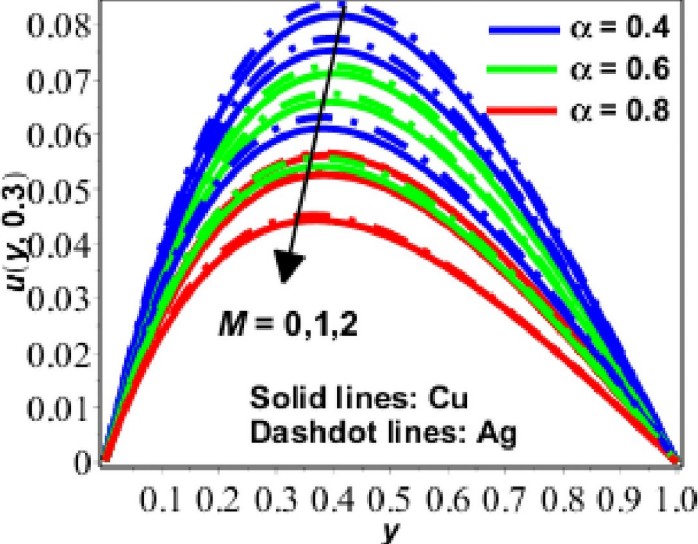

**Fig 1. Variation in $u(y, t)$ against $M$ for Pr = 6.2, $Gr$ = 0.9, $N$ = 0.9, $\phi$ = 0.2, $R$ = 0.9.**

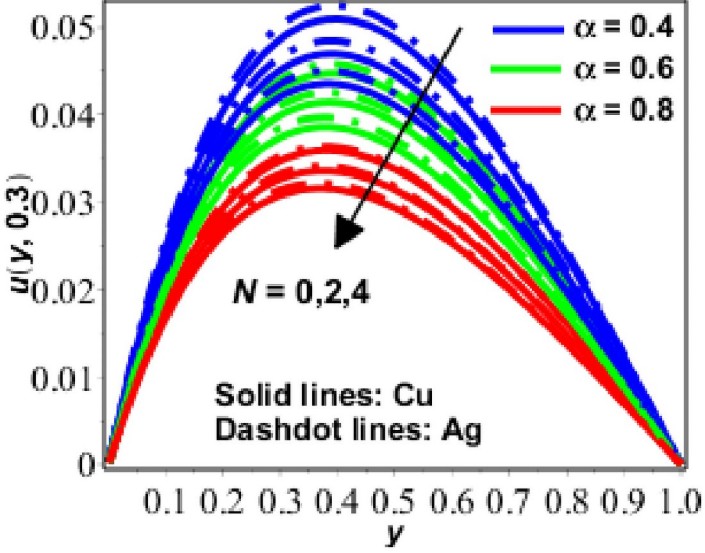

**Fig 2. Variation in $u(y, t)$ against $N$ for Pr = 6.2, $Gr$ = 0.9, $M$ = 0.9, $\phi$ = 0.2, $R$ = 0.9.**

The effect of Gr on flow velocity is seen in Fig 3, which demonstrates that velocity increases as the size of *Gr* increases. As seen in the preceding section, the Grashof number is inversely related to the viscosity $\mu$. Increased values of the Grashof number indicate that the viscosity is reducing, which explains why the velocity function is growing. Generally, an increase in any buoyancy-related parameter such as the Grashof number in the present case, causes an increase in the wall temperature, which weakens the bond(s) between the fluids, reduces internal friction pressure, and makes gravity stronger (i.e. makes the specific weight appreciably different between the immediate fluid layers adjacent to the wall). For detailed analysis of

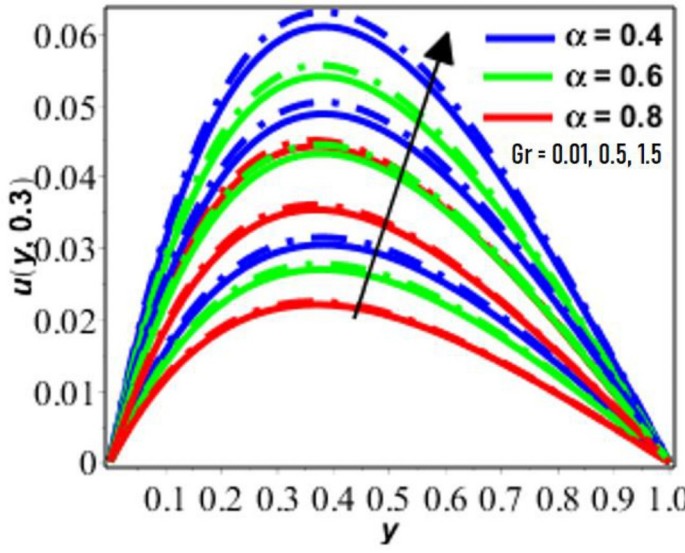

**Fig 3. Variation in $u(y, t)$ against $Gr$ for Pr = 6.2 $R$ = 0.9, $M$ = 2, $N$ = 0.2, $\phi$ = 0.2.**

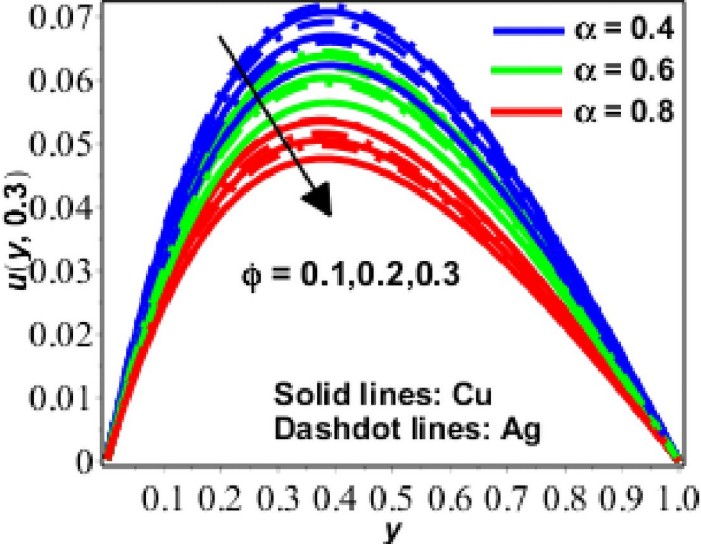

**Fig 4. Variation in $u(y, t)$ against $R$ for Pr = 6.2 $R = 0.5$, $Gr = 0.5$, $M = 2$, $N = 0.9$.**

Grashof number and its effect on the fluid motion, three different scenarios (transport phenomenon) are discussed, namely: when (a) Grashof number is greater than 1, (b) Grashof number is less than 1, and (c) Grashof number is small (Gr = 0.01). In this first case, it is observed that velocity increases with increasing value of temperature-dependent viscosity parameter when Grashof number is greater than 1. In the second case, the observation showed that velocity decreases with increasing values of temperature-dependent viscosity parameter when Grashof number is less than 1. The third case examines the flow situation when the

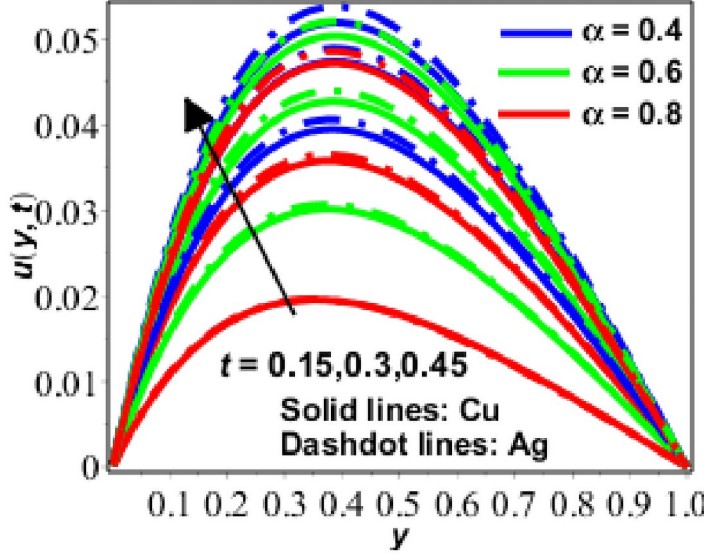

**Fig 5. Variation in $u(y, t)$ against $t$ for $Gr = 0.5$ $R = 0.5$, $M = 2$, $N = 0.9$, $\phi = 0.2$, Pr = 6.2.**

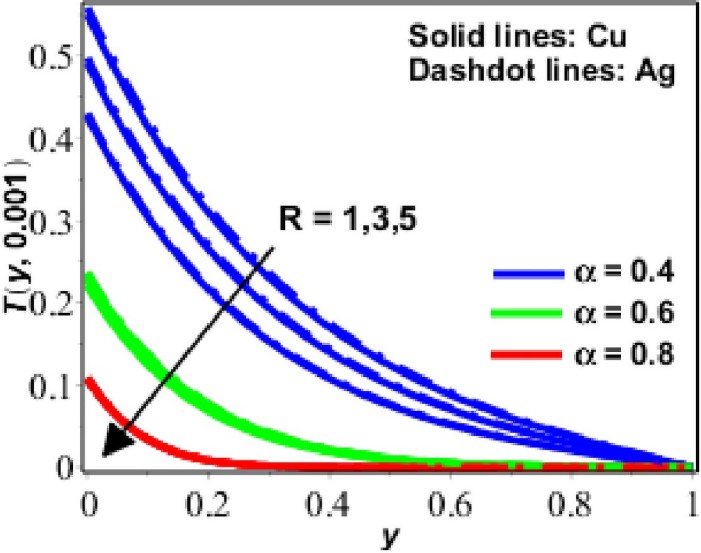

**Fig 6. Behavior of *T*(*y*, *t*) against *R* for Pr = 6.2 *ϕ* = 0.1.**

when Grashof number is moderately small i.e. Gr = 0.01. The above observation shows that the Grashof number in convection flow plays and important role (Fig 3) [44,45].

Volume fraction is often used in solid materials science and engineering to refer to the concentration of a given phase, that is, the ratio of the volume of the particular phase to the total volume of the sample. Here in our study, *ϕ* shows the solid volume fraction of nanofluid. Additionally, Fig 4 has been displayed for various numerical values. As seen in Fig 4, velocity decreases as the numerical value of for nanofluid increases. In Fig 5, the velocity behaviour for changing t has been illustrated, demonstrating that flow velocity steadily increases with time.

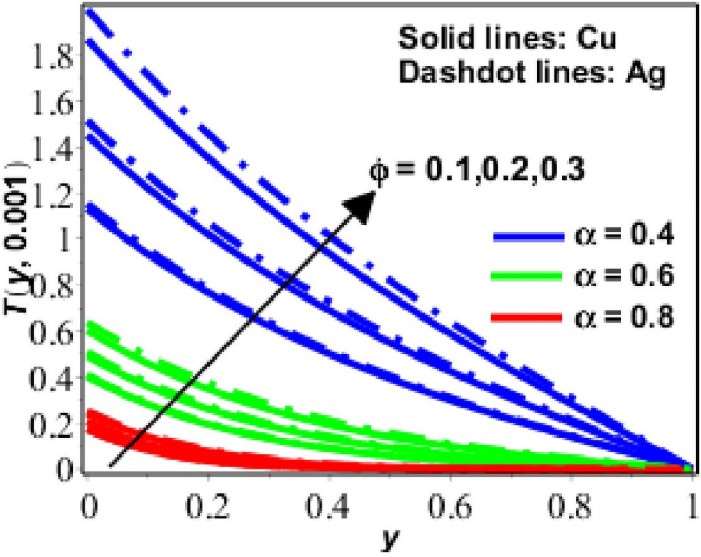

**Fig 7. Behavior of *T*(*y*, *t*) against *ϕ* for Pr = 6.2.**

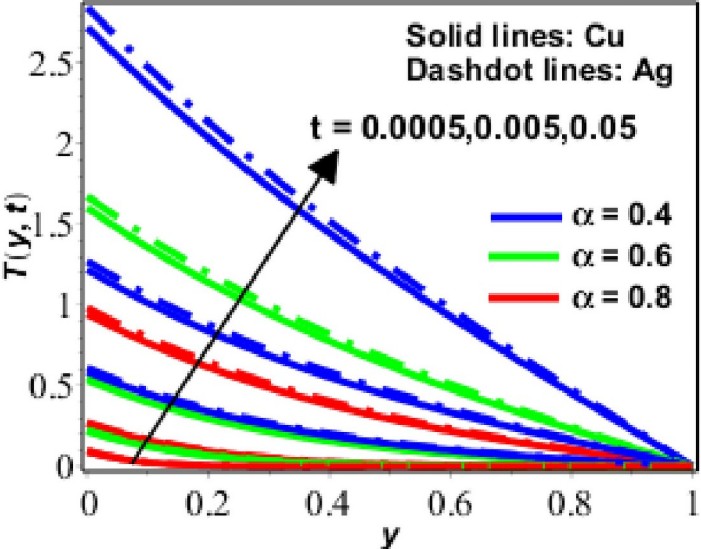

**Fig 8. Variation in $T(y, t)$ against $t$ for $R = 0.1$ $\phi = 0.1$, Pr = 6.2.**

The temperature performance of nanofluids (based on copper ($Cu$) and silver ($Ag$)) has been presented in Figs 6–9 for various numerical parameter values and fractional parameter values. Fig 6 demonstrates that the temperature of the nanofluid decreases as the radiation effects $N$ increase, indicating that the nanofluid is radiative in nature and radiates energy. As a result of this, radiation (in the form of electromagnetic waves) wastes energy, reducing the thickness of the thermal boundary layer and therefore the temperature. The temperature increases as the value of the solid volume fraction ($\phi$) disclosed in Fig 7 which is increased. In Fig 8, a similar response of the temperature of the nanofluid was seen when the numerical

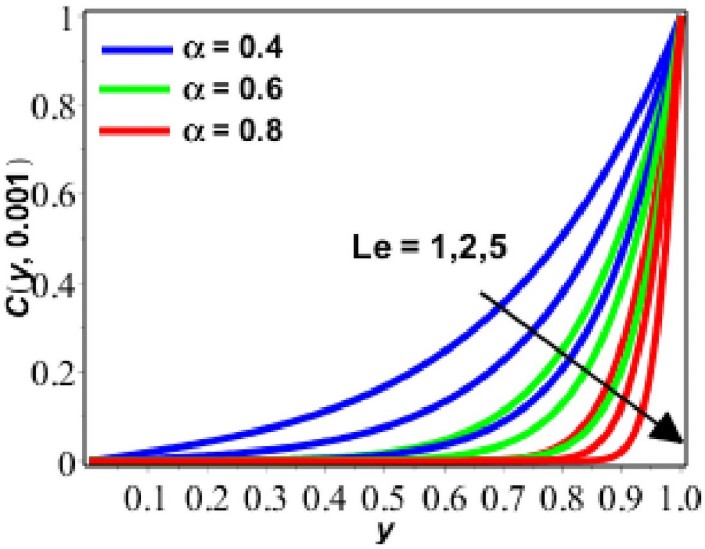

**Fig 9. Behavior of $C(y, t)$ against $Le$ for $\delta = 0.9$.**

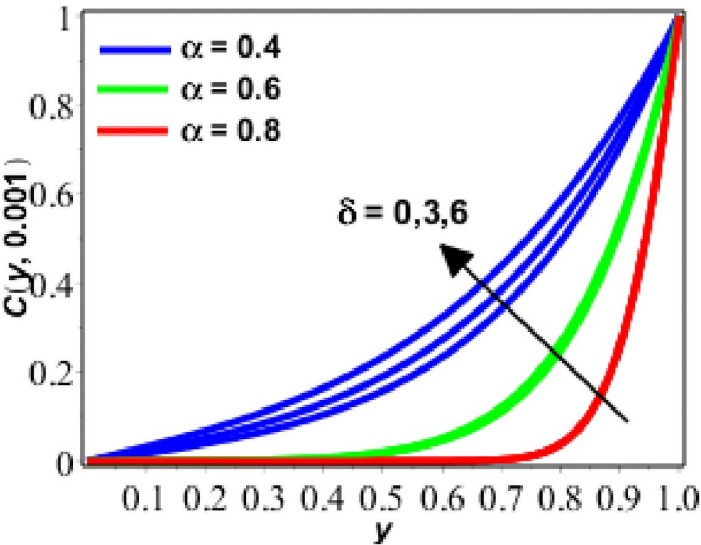

**Fig 10. Variation in $C(y, t)$ against $\delta$ for $Le = 1$.**

values $t$ and $\alpha$ were changed. Moreover, as seen in Fig 8, nano fluids based on silver have a greater temperature than nano fluids based on copper.

Enactment of concentration of solute is presented on Figs 9–11 for diverse values of parameters. Figs 9 and 10 show the performance of concentration of solute for upsurging numerical values of Lewis number ($Le$) and $\delta$. Fig 9 shows that concentration and concentration boundary layer thickness is decreeing as we are increasing the magnitude of $Le$. The motive behind is that when the $Le$ increases, the diffusion process decreases because of the inversely proportional relationship between $Le$ and diffusion. As the diffusion process decreases, the

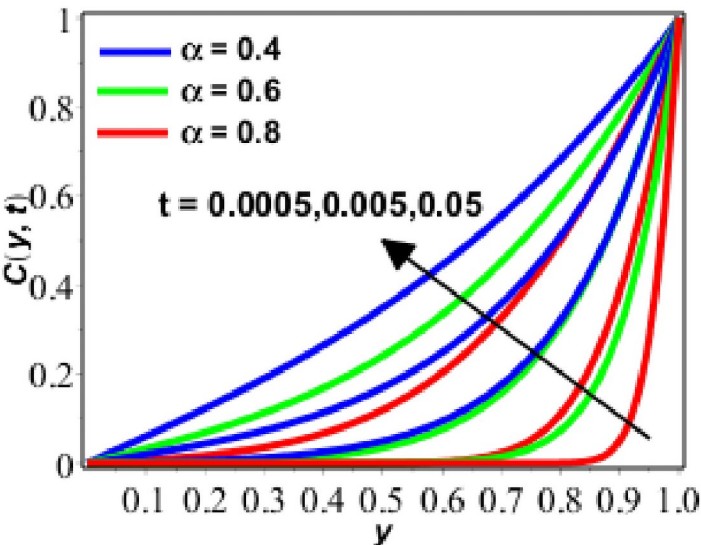

**Fig 11. Behavior of $C(y, t)$ against $t$ for $\delta = 0.9$ $Le = 2$.**

**Table 1. The expressions of parameters used in Eqs (10)–(12) are given in tabular form.**

| Parameter | Expression | Parameter | Expression |
|---|---|---|---|
| Grashof number | $Gr = \frac{g\beta_f\left(T_w - T_\infty\right)d^3}{v_f^2}$ | Prandtl number | $\Pr = \frac{\mu_f(c_p)_f}{k_f}$ |
| Hartmann number | $M^2 = \frac{\sigma B_0^2 d^2}{\mu_f}$ | Radiation | $R = \frac{4Id^2}{k_f}$ |
| Porosity | $N = \frac{d^2\phi_m}{k_f}$ | Lewis number | $Le = \frac{v_f}{D}$ |
| Reaction rate | $\delta = \frac{\gamma d^2\left(C_w - C_\infty\right)^{n-1}}{v_f}$ | Fractional | $\alpha$ |

concentration of solute is also decreased. On the other hand, the response rate parameter exhibits the inverse relationship. In this situation, we can readily assert that the rate of chemical reaction rises as the reaction rate parameter increases. This becomes the cause for the growing behaviour of the solute's concentration. In Fig 10, the concentration of solute increases with increasing $\delta$ and decreases with increasing fractional parameter. In the last Fig 11, the concentration of solute increases with the passage of time and with the decreasing values of the fractional parameter $(\alpha)$.

## 5. Conclusion

The viscous, incompressible, and convection-free fluidic flow near an isothermal vertical plate is theoretically investigated in this article. The plate's velocity varies with time, and its motion is translational. The fractional differential equations are used to express the governing flow equations (FDEs). A mixture of finite difference and Crank Nicolson techniques is used to find numerical solutions for solute concentration, velocity, and temperature. As a result, the following is the main summary of our research:

- The outlines of temperature, velocity, and concentration reduced, while the total numerical values of parameter $\alpha$ decreased.

- Values for velocity and temperature for the case of Ag-based nanofluid is more than Cu-based nanofluid

- Ordinary fluid flow is more leisurely than fractional fluid flow.

- The velocity, temperature, and concentration profiles all showed declining behaviour as time passed.

## Supporting information

**S1 Abbreviations.**
(DOCX)

## Author Contributions

**Conceptualization:** Tamour Zubair, Madiha Ghamkhar.

**Data curation:** Muhammad Usman, Kottakkaran Sooppy Nisar.

**Formal analysis:** Ilyas Khan, Muhammad Ahmad.

**Investigation:** Tamour Zubair, Muhammad Usman, Muhammad Ahmad.

**Methodology:** Muhammad Usman.

**Resources:** Kottakkaran Sooppy Nisar.

**Software:** Kottakkaran Sooppy Nisar.

**Validation:** Tamour Zubair, Ilyas Khan, Muhammad Ahmad.

**Visualization:** Muhammad Usman.

**Writing – review & editing:** Tamour Zubair.

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
