## [Editor Report · Decision Letter 0]

5 Oct 2021

PONE-D-21-31025Crank Nicholson Scheme to Examine the Fractional-order Unsteady Nanofluid Flow of Free Convection of Viscous FluidsPLOS ONE

Dear Dr. Zubair,

Thank you for submitting your manuscript to PLOS ONE. After careful consideration, we feel that it has merit but does not fully meet PLOS ONE’s publication criteria as it currently stands. Therefore, we invite you to submit a revised version of the manuscript that addresses the points raised during the review process.

We look forward to receiving your revised manuscript.

Kind regards,

Academic Editor

PLOS ONE

Journal Requirements:

2. Please update your submission to use the PLOS LaTeX template. The template and more information on our requirements for LaTeX submissions can be found at http://journals.plos.org/plosone/s/latex

Additional Editor Comments (if provided):

Authors must address the following issues before considering this paper for review.

1. Strengthen the mathematical modelling of the problem.

2. Update the introduction with relevant references only. Also arrange the references in proper format.

3.Validate the results.
---

## [Author Response · Author response to Decision Letter 0]

18 Oct 2021

Response to reviewer’s comments

Manuscript ID: PONE-D-21-31025 

Journal Name: PLOS ONE

Dear Editor-in-Chief,

Thank you for your useful comments on our manuscript. We have considered your additional editorial comments and made the following changes in our paper entitled “Crank Nicholson Scheme to Examine the Fractional-order Unsteady Nanofluid Flow of Free Convection of Viscous Fluids". We have modified the manuscript accordingly, and detailed corrections are listed below point by point. All the changes of are highlighted in red colour.

Additional Editor Comments & Response:

1. Strengthen the mathematical modelling of the problem.

Response: Geometry of the problem has been added to fix the query.

2. Update the introduction with relevant references only. Also arrange the references in proper format.

Response: Introduction section has been updated according to the instructions. The format of the references has been improved.

3. Validate the results.

Response: Graphical results and further analysis on the base of presented figures has been added.

 With Regards, 

 Tamour Zubair

---

## [Decision Letter · Decision Letter 1]

25 Oct 2021

PONE-D-21-31025R1Crank Nicholson Scheme to Examine the Fractional-order Unsteady Nanofluid Flow of Free Convection of Viscous FluidsPLOS ONE

Dear Dr. Zubair,

Thank you for submitting your manuscript to PLOS ONE. After careful consideration, we feel that it has merit but does not fully meet PLOS ONE’s publication criteria as it currently stands. Therefore, we invite you to submit a revised version of the manuscript that addresses the points raised during the review process.

We look forward to receiving your revised manuscript.

Kind regards,

Academic Editor

PLOS ONE

Reviewers' comments:

Reviewer's Responses to Questions

6. Review Comments to the Author

Reviewer #1: The manuscript presents the outcome of a research study on the fractional-order of unsteady

nanofluid flow due to free convection using Crank Nicholson integration scheme. The contribution of the report to the body of knowledge is significant and novel. Also, the aim and objectives of the study are within the scope of PLOS ONE. However, the present form of the report needs revision. The author should consider the following points:

Q1. The title is wrongly written. It was written, "Crank Nicholson Scheme to Examine the Fractional-order Unsteady

Nanofluid Flow of Free Convection of Viscous Fluids". Meanwhile, the report actually presents, "fractional-order of unsteady

nanofluid flow due to free convection using Crank Nicholson integration scheme"

Comment: Revise the title or consider, "Analysis of fractional-order of unsteady nanofluid flow due to free convection using Crank Nicholson integration scheme."

Q2. An effective Abstract is a global paragraph that prepares the reader for the rest of the document. The present form of the abstract is not acceptable because it is not appropriate.

Comment 1: Use

(1) one sentence to present the significance of the study,

(2) one sentence to present the aim of the study,

(3) one sentence to present the research methodology, and

(4) two sentences to present the conclusion drawn from the study.

Comment 2: Do not use the pronoun, "we" in a scientific report like this. Also, the first sentence of the abstract should focus on the significance of the study on, "Analysis of fractional-order of unsteady nanofluid flow due to free convection using Crank Nicholson integration scheme"

Q3. The present form of the introduction lacks structure and strongly rejected. It is important to release two comments which are

Comment 1: Writing is constructed by putting sentences in sequence, one after another. Meaning should flow from one sentence to the next, carrying the argument or point of view forward clearly and concisely.

Comment 2: Use the style mentioned above to revise not only the introduction section but also the analysis and discussion of results. This is necessary because some paragraphs are somehow scanty and not structure.

Comment 3:

After reading the introduction section, it is worth noticing that the present form of the theoretical and empirical reviews seems okay but not connecting related facts to open a gap which this report intends to fill.

Comment 4: Revise. The authors may check the following videos for insight: https://youtu.be/ugFJnflnsF0

https://youtu.be/1A27y6eUsqA

https://youtu.be/5GiT-N9Y_Ig

Q4. Revise the introduction. Update with scientific facts like:

Q4a) The manuscript lacks important facts on the choice of the value for Prandtl number.

Comment: Update the manuscript with the fact that "…the appropriate Prandtl number for (a) Methanol based nanofluid is 7.3786, (b) water-based nanofluid is 6.1723, (c) blood-based nanofluid is 22.9540, and Ethylene Glycol based nanofluid is 150.46."

Comment: Cite the Source - Physica Scripta 95(9), 095205. https://doi.org/10.1088/1402-4896/aba8c6

Q5. Update the discussion of the results illustrated as Fig. 13 and Fig. 14 with the fact that

a) An increase in the Grashof number or some other buoyancy-related parameter means an increase in the wall temperature, which weakens the bond(s) between the fluids, reduces internal friction pressure, and makes gravity stronger (i.e. makes the specific weight appreciably different between the immediate fluid layers adjacent to the wall).

You are expected to revise the fact above using your group of words.

Cite the source: Journal of Molecular Liquids 249, 980 – 990, 2018. https://doi.org/10.1016/j.molliq.2017.11.042

Q6. The analysis of the effect of Grashof number is not logical and insightful. Permit me to remind you of heating the surface and cooling the surface.

Comment: Do you know that there are three different implications of increasing Grashof number proportional to the buoyancy force? Could you try to extend your simulation to check what will happen to the transport phenomenon when (a) Grashof number is greater than 1, (b) Grashof number is less than 1, and (c) Grashof number is small like 0.01. Relate your observation(s) with the first conclusion (i.e. Velocity increases with increasing value of temperature-dependent viscosity parameter when Grashof number is greater than 1; velocity decreases with increasing values of temperature-dependent viscosity parameter when Grashof number is less than 1. The two effects exist when Grashof number is moderately small) reached in page 117 of Open Journal of Fluid Dynamics, 2015, 5, 106-120. http://dx.doi.org/10.4236/ojfd.2015.52013.

Q7. Section three is loaded with new results that are not clear. Nothing is known on the direction of thought. Research Questions are needed at the end of the introduction to harmonize the findings of the study.

Comment 1: The video below may be useful to guide the authors

https://youtu.be/cn4iRvtwf8M

Comment 2: The author should update the manuscript with appropriate and relevant research questions at the end of the introduction section. This would guide the author to structure a logical analysis of results. Logical questions are expected. This would help readers to link what is known in the literature with the novelty of this study. Samples of Research Questions can be found in the following reports

A. https://doi.org/10.1038/s41598-021-81417-y

B. https://doi.org/10.1007/s10973-021-10550-7

Reviewer #2: 

 <table border="0" class="datatable" id="ctlReviewQuestionsResponses_AnswersMatrix" style="border: 1px solid rgb(204, 204, 204); border-collapse: collapse; width: 1256.52px; line-height: 14px; color: rgb(0, 0, 51); font-family: verdana, geneva, arial, helvetica, sans-serif; font-size: 11.2px; background-color: rgb(244, 244, 244);"> 

1. Write at least two sentences that highlights the practical application of this research.

2. Paper should be carefully revised for punctuation, grammar, space, and spelling mistakes. There are number of such kind of mistakes that must be tackled.

3. Nomenclature can help the readers to understand the number of variables used in the article. So add a nomenclature with the units.

4. Physical reasoning is missing in most of figures. Write in detail the true physical reason for increasing or decreasing.

5. The introduction section needs to be carefully revised. It is suggested that most related articles should be cited. The following articles can help you to improve your introduction section.

https://doi.org/10.1016/j.icheatmasstransfer.2021.105563

https://doi.org/10.1177/09544089211043605

https://doi.org/10.1002/htj.22280

https://doi.org/10.1002/htj.21947

https://doi.org/10.1007/s10973-020-09943-x

https://doi.org/10.1080/01430750.2020.1831593

https://doi.org/10.1038/s41598-020-61215-8

6. The abstract should be updated with fruitful outcomes of the problem.

7. All governing equations need to be referenced properly.

8. Some unclear words are used in the paper rectify it.(less...)

</table>**********

7. PLOS authors have the option to publish the peer review history of their article (what does this mean?). If published, this will include your full peer review and any attached files.

---

## [Author Response · Author response to Decision Letter 1]

9 Dec 2021

Response to Reviewer’s Comments

Manuscript ID: PONE-D-21-31025R1 

Journal Name: Plos One

Dear Editor-in-Chief,

Thank you for your useful comments on our manuscript. We have considered your editorial or reviewer's comments and made the following changes in our paper entitled “Analysis of Fractional-order of Unsteady Nanofluid Flow Due to Free Convection Using Crank Nicholson Integration Scheme ”. We have modified the manuscript accordingly, and detailed corrections are listed below point by point. All the changes of referee-I and referee-II is highlighted in red and purple colors respectively.

Comments from Reviewers-I & Response:

1. The title is wrongly written. It was written, "Crank Nicholson Scheme to Examine the Fractional-order Unsteady Nanofluid Flow of Free Convection of Viscous Fluids". Meanwhile, the report actually presents, "fractional-order of unsteady nanofluid flow due to free convection using Crank Nicholson integration scheme" 

Comment: Revise the title or consider, "Analysis of fractional-order of unsteady nanofluid flow due to free convection using Crank Nicholson integration scheme."

Response: Revised as suggested.

2. An effective Abstract is a global paragraph that prepares the reader for the rest of the document. The present form of the abstract is not acceptable because it is not appropriate. 

Comment 1: Use (1) one sentence to present the significance of the study, (2) one sentence to present the aim of the study, (3) one sentence to present the research methodology, and (4) two sentences to present the conclusion drawn from the study.

Comment 2: Do not use the pronoun, "we" in a scientific report like this. Also, the first sentence of the abstract should focus on the significance of the study on, "Analysis of fractional-order of unsteady nanofluid flow due to free convection using Crank Nicholson integration scheme"

Response: Revised as suggested.

3. The present form of the introduction lacks structure and strongly rejected. It is important to release two comments which are 

Comment 1: Writing is constructed by putting sentences in sequence, one after another. Meaning should flow from one sentence to the next, carrying the argument or point of view forward clearly and concisely. 

Comment 2: Use the style mentioned above to revise not only the introduction section but also the analysis and discussion of results. This is necessary because some paragraphs are somehow scanty and not structure. 

Comment 3: After reading the introduction section, it is worth noticing that the present form of the theoretical and empirical reviews seems okay but not connecting related facts to open a gap which this report intends to fill. 

Comment 4: Revise. The authors may check the following videos for insight: https://youtu.be/ugFJnflnsF0
https://youtu.be/1A27y6eUsqA
https://youtu.be/5GiT-N9Y_Ig

Response: Response: Revised as suggested. All the four comments are included.

4. Revise the introduction. Update with scientific facts like: 

a) The manuscript lacks important facts on the choice of the value for Prandtl number. 

Comment: Update the manuscript with the fact that "…the appropriate Prandtl number for (a) Methanol based nanofluid is 7.3786, (b) water-based nanofluid is 6.1723, (c) blood-based nanofluid is 22.9540, and Ethylene Glycol based nanofluid is 150.46." 

Comment: Cite the Source - Physica Scripta 95(9), 095205. https://doi.org/10.1088/1402-4896/aba8c6

Response: As we have considered viscous fluid model, therefore, the base fluid is limited to water only with Prandtl number as 6.2. This correction is done in all figures, and the figure for variation of Prandtl number is deleted. All calculations are now done for water-based nanofluid with Pr=6.2, with its fixed value. The suggested reference is also included in the revised manuscript, see Ref. [44].

5. Update the discussion of the results illustrated as Fig. 13 and Fig. 14 with the fact that a) An increase in the Grashof number or some other buoyancy-related parameter means an increase in the wall temperature, which weakens the bond(s) between the fluids, reduces internal friction pressure, and makes gravity stronger (i.e. makes the specific weight appreciably different between the immediate fluid layers adjacent to the wall). You are expected to revise the fact above using your group of words. Cite the source: Journal of Molecular Liquids 249, 980 – 990, 2018. https://doi.org/10.1016/j.molliq.2017.11.042

Response: Dear referee, in this manuscript we don’t have Fig. 13 and Fig. 14, however, the suggested correction is done in Fig. 3, as this include the Grashof number related information. The suggested reference is also included in the revised manuscript, see Ref. [45].

6. The analysis of the effect of Grashof number is not logical and insightful. Permit me to remind you of heating the surface and cooling the surface. 

Comment: Do you know that there are three different implications of increasing Grashof number proportional to the buoyancy force? Could you try to extend your simulation to check what will happen to the transport phenomenon when (a) Grashof number is greater than 1, (b) Grashof number is less than 1, and (c) Grashof number is small like 0.01. Relate your observation(s) with the first conclusion (i.e. Velocity increases with increasing value of temperature-dependent viscosity parameter when Grashof number is greater than 1; velocity decreases with increasing values of temperature-dependent viscosity parameter when Grashof number is less than 1. The two effects exist when Grashof number is moderately small) reached in page 117 of Open Journal of Fluid Dynamics, 2015, 5, 106-120. http://dx.doi.org/10.4236/ojfd.2015.52013.

Response: Dear referee, the above suggestions are included, and the authors are thankful for such a useful suggestion. The manuscript is revised accordingly, please refer to the discussion of Fig, 3. The suggested reference is also included in the revised manuscript, see Ref. [46].

7. Section three is loaded with new results that are not clear. Nothing is known on the direction of thought. Research Questions are needed at the end of the introduction to harmonize the findings of the study. 

Comment 1: The video below may be useful to guide the authors https://youtu.be/cn4iRvtwf8M Comment 2: The author should update the manuscript with appropriate and relevant research questions at the end of the introduction section. This would guide the author to structure a logical analysis of results. Logical questions are expected. This would help readers to link what is known in the literature with the novelty of this study. Samples of Research Questions can be found in the following reports A. https://doi.org/10.1038/s41598-021-81417-y B. https://doi.org/10.1007/s10973-021-10550-7

Response: The manuscript is revised accordingly. The two suggested report are also included please refer to Res. [47] and [48]. 

Comments from Reviewers-II & Response:

1. Write at least two sentences that highlights the practical application of this research.

Response: Our study is very significant and is applicable in different area of science as follow. 

• It is applicable in antibacterial study.

• Effects of shape control is applicable in inorganic materials.

• The study of nano-particles in term are applicable in biomedical.

All the relevant applications (along with references) of our study are updated in the paper and suitable references are also added. 

2. Paper should be carefully revised for punctuation, grammar, space, and spelling mistakes. There are number of such kind of mistakes that must be tackled.

Response: The manuscript revised carefully and all mistakes are removed.

3. Nomenclature can help the readers to understand the number of variables used in the article. So add a nomenclature with the units.

Response: Nomenclature is added.

4. Physical reasoning is missing in most of figures. Write in detail the true physical reason for increasing or decreasing. 

Response: Result and discussion section has been improved with physical reasons.

5. The introduction section needs to be carefully revised. It is suggested that most related articles should be cited. The following articles can help you to improve your introduction section. https://doi.org/10.1016/j.icheatmasstransfer.2021.105563
https://doi.org/10.1177/09544089211043605
https://doi.org/10.1002/htj.22280
https://doi.org/10.1002/htj.21947
https://doi.org/10.1007/s10973-020-09943-x
https://doi.org/10.1080/01430750.2020.1831593
https://doi.org/10.1038/s41598-020-61215-8

Response: The references have been cited at suitable positions. 

6. The abstract should be updated with fruitful outcomes of the problem.

Response: The abstract has been improved.

7. All governing equations need to be referenced properly.

Response: All governing equations have been cited accordingly.

8. Some unclear words are used in the paper rectify it.

Response: Done as suggested.

 With Regards, 

 Tamour Zubair

---

## [Decision Letter · Decision Letter 2]

13 Dec 2021

Crank Nicholson Scheme to Examine the Fractional-order Unsteady Nanofluid Flow of Free Convection of Viscous Fluids

PONE-D-21-31025R2

Dear Dr. Zubair,

We’re pleased to inform you that your manuscript has been judged scientifically suitable for publication and will be formally accepted for publication once it meets all outstanding technical requirements.

Kind regards,

Naramgari Sandeep, Ph.D

Academic Editor

PLOS ONE

Reviewers' comments:

Reviewer's Responses to Questions

6. Review Comments to the Author

Reviewer #1: Checking through the revised version, it is worth mentioning that

a. the manuscript contains an interesting and novel aim,

b. the title is informative and relevant,

c. the introduction, literature review, methodology, results, discussion of results, conclusion and references are of high standard,

d. Author(s) have rigorously revised the manuscript. The present form of the whole report is also of high standard, and

e. the contribution of the report to the body of knowledge is significant.

Based on these aforementioned facts, it is worth concluding that the article is error free and suitable for publication. I hereby recommend "Acceptance".

---

## [Editor Report · Acceptance letter]

7 Jan 2022

PONE-D-21-31025R2 

Crank Nicholson Scheme to Examine the Fractional-order  Unsteady Nanofluid Flow of Free Convection of Viscous Fluids 

Dear Dr. Zubair:

I'm pleased to inform you that your manuscript has been deemed suitable for publication in PLOS ONE. Congratulations! Your manuscript is now with our production department. 

Kind regards, 

on behalf of

Dr. Naramgari Sandeep 

Academic Editor

PLOS ONE